# LongLive: Real-time Interactive Long Video Generation

**Shuai Yang**[1,3]  **Wei Huang**[1,4]  **Ruihang Chu**[5]  **Yicheng Xiao**[5]  **Yuyang Zhao**[1]

**Xianbang Wang**[2]  **Muyang Li**[2]  **Enze Xie**[1]  **Yingcong Chen**[3]  **Yao Lu**[1]

**Song Han**[1,2]  **Yukang Chen**[1]

[1]**NVIDIA**  [2]**MIT**  [3]**HKUST(GZ)**  [4]**HKU**  [5]**THU**

## Abstract

We present LongLive, a frame-level autoregressive (AR) framework for real-time and interactive long video generation. Long video generation presents challenges in both efficiency and quality. Diffusion and Diffusion-Forcing models can produce high-quality videos but suffer from low efficiency due to bidirectional attention. Causal attention AR models support KV caching for faster inference, but often degrade in quality on long videos due to memory challenges during long-video training. In addition, beyond static prompt-based generation, interactive capabilities, such as streaming prompt inputs, are critical for dynamic content creation, enabling users to guide narratives in real time. This interactive requirement significantly increases complexity, especially in ensuring visual consistency and semantic coherence during prompt transitions. To address these challenges, LongLive adopts a causal, frame-level AR design that integrates a ***KV-recache*** mechanism that refreshes cached states with new prompts for smooth, adherent switches; ***streaming long tuning*** to enable long video training and to align training and inference (train-long–test-long); and ***short window attention*** paired with a ***frame-level attention sink***, shorten as frame sink, preserving long-range consistency while enabling faster generation. With these key designs, LongLive fine-tunes a 1.3B-parameter short-clip model to minute-long generation in just 32 GPU-days. At inference, LongLive sustains 20.7 FPS on a single NVIDIA H100, achieves strong performance on VBench in both short and long videos. LongLive supports up to 240-second videos on a single H100 GPU. LongLive further supports INT8-quantized inference with only marginal quality loss. our code is available at: https://github.com/NVlabs/LongLive

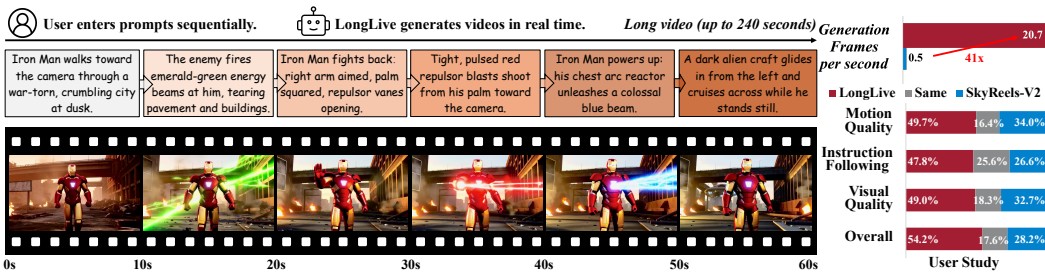

Figure 1: The workflow of LongLive. LongLive accepts sequential user prompts and generates corresponding videos in real time, enabling user-guided long video generation. The 60-second sequence shown is an example, LongLive supports up to 240-second videos in a single H100 GPU.

## 1 Introduction

Long video generation is essential for advancing creative, educational, and cinematic applications. It enables coherent storytelling, richer scene development, and more complex temporal dynamics

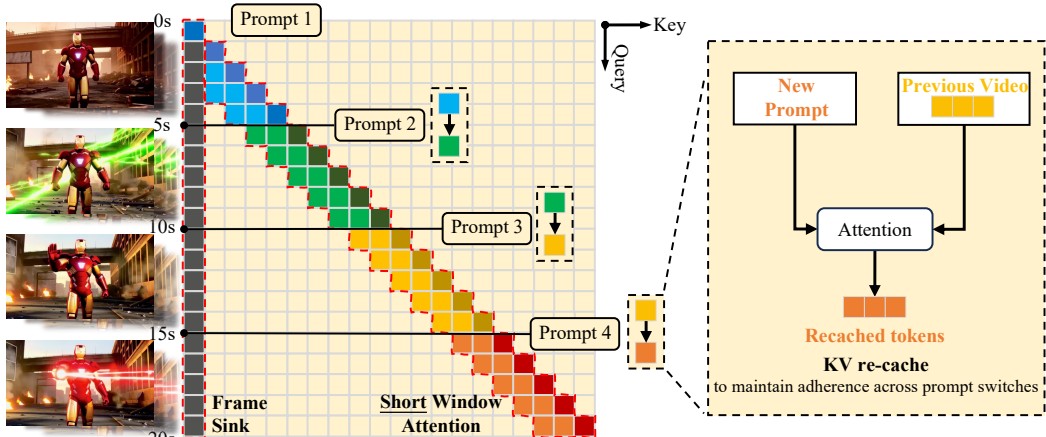

Figure 2: The framework of LONGLIVE. (Left) LONGLIVE processes sequential user prompts and generates a corresponding long video using efficient **short window attention** and **frame sink**. Compared to the normal attention window of 5s, our short window only uses half the size, with the help of frame sink, which maintains the long-range consistency. (Right) To maintain consistency when the prompt switches, LONGLIVE employs a **KV-recache** technique that updates cached key–value states by combining previous videos with new prompt embeddings through cross-attention layers.

than short clips can provide. However, static prompt-based generation limits adaptability once the process has commenced. It is difficult for users to conceive highly detailed, long-form prompts in a single step. Beyond simply producing long videos, the ability to interact alongside the generation process, such as streaming prompt inputs during runtime, opens new possibilities for adaptive content creation. This interactive paradigm enables users to guide narratives, adjust visual styles, or introduce new elements on the fly. Therefore, interaction makes long video generation controllable.

Interactive long video generation poses difficulties in both quality and efficiency. For the *quality* perspective, it is difficult to maintain smooth, consistent, and coherent transitions when switching between user prompts during generation. Even subtle mismatches in visual style, motion continuity, or scene layout can disrupt the narrative flow and reduce the overall realism of the video. For the *efficiency* perspective, the computational and memory demands scale rapidly with video length. For example, generating a 180-second video with the Wan-2.1 (Wan et al., 2025) model requires processing over one million tokens, which is computationally prohibitive. In addition, in an interactive setting, prolonged user waiting times severely degrade the overall user experience.

Existing video generation methods have limitations in long video generation. For *diffusion-based* video generation models (Wan et al., 2025; Kong et al., 2024; Yang et al., 2025; Wei et al., 2025; OpenAI, 2024; Kuaishou, 2024) and *diffusion-forcing* models (Chen et al., 2024a; 2025a; Zhang & Agrawala, 2025), although they can produce high-quality short clips, their reliance on bidirectional attention makes inference inefficient. The bidirectional attention prevents KV (key–value) cache technique, leading to redundant computation and prohibitive latency for long videos. For example, SkyReels-V2 (Chen et al., 2025a) requires approximately 50 minutes on an H100 GPU to generate a 60-second video. For *AR* models with causal attention, they can leverage cached KV states for faster inference, but they often exhibit degraded quality when generating long videos. Due to the high cost of directly training on long videos, existing AR models (Huang et al., 2025; Teng et al., 2025) typically adopt a train-short-test-long strategy. Consequently, the quality gradually degrades as the video length increases. In the interactive setting involving prompt switching, error accumulation, and loss of temporal coherence over time further result in visual artifacts and inconsistency.

In this paper, we propose LONGLIVE, a real-time interactive long video generation framework, as illustrated in Figure 1. LONGLIVE is a causal attention, frame-level AR video generation model, enabling it to inherit the KV cache mechanism for efficient inference. Our key design is ***KV-recache***, as shown in Figure 2, which updates cached states by incorporating new prompt embeddings. This technique ensures both smoothness and prompt adherence across prompt switches in interactive settings. In addition, for efficient fine-tuning, we present a ***streaming long tuning*** strategy that

preserves consistency between training and inference (train-long-test-long), to address the degradation commonly observed in long-video AR generation. For efficient inference, we introduce ***short window attention*** combined with a ***frame-level attention sink*** (abbreviated as frame sink), which significantly accelerates inference while preserving performance.

In our experiments, LONGLIVE delivers both high efficiency and strong quality for interactive long-video generation. In terms of training efficiency, we fine-tune a 1.3B-parameter model to produce high-quality minute-long videos in only 32 GPU-days. Training on long videos is essential: it not only improves long-horizon fidelity but also enables efficient inference strategies that markedly accelerate decoding. In terms of inference efficiency, LONGLIVE sustains 20.7 FPS on a single NVIDIA H100, supporting real-time interaction and outperforming state-of-the-art approaches in throughput. In terms of quality, our framework achieves strong VBench scores on both short- and long-video settings. LONGLIVE scales to produce videos up to 240 seconds, on a single H100 GPU, while maintaining high visual fidelity and temporal coherence, effectively handling long video generation with little degradation. Moreover, we further enable INT8-quantized inference in LONGLIVE, with only marginal quality loss, as shown in Appendix H.

## 2 RELATED WORK

We present core related work here and provide extended discussion with details in the appendix D. A growing number of works (Chen et al., 2024a; Song et al., 2025; Mao et al., 2025; Yuan et al., 2025; Zhang & Agrawala, 2025; Gao et al., 2025; Henschel et al., 2025) integrate diffusion modeling with AR prediction, an intermediate paradigm between purely diffusion-based approaches and purely AR approaches. SkyReels-V2 (Chen et al., 2025a) couples diffusion forcing with a film-structure planner and multimodal controls. Recent efforts (Yin et al., 2025; Huang et al., 2025; Gu et al., 2025; Teng et al., 2025) have advanced causal AR-based models for long video generation. StreamDiT (Kodaira et al., 2025) trains a diffusion model with window attention, but has potential drift or detail loss over long streams. AAPT (Lin et al., 2025) transforms a pre-trained latent video diffusion model into a 1-NFE, real-time streaming generator using adversarial post-training and supports interactive control via camera and human pose, whereas *LongLive* instead uses diffusion-based distribution-matching and train-long-test-long distillation for text-driven, multi-minute generation without GAN post-training. Most recently, Self-forcing (Huang et al., 2025) addresses the train–test gap in AR video diffusion by simulating inference conditions during training, rolling out generation with KV cache, and conditioning on model outputs. MAGI-1 (Teng et al., 2025) scales AR video generation to large models and datasets through chunk-wise prediction, but its prompt switching requires manual adjustment of KV-cache windows at different steps.

## 3 METHOD

### 3.1 KV RECACHE

Causal AR models naturally support interactive prompt switching, but this ability is limited. Discarding all prior KV cache at the switch improves adherence to the new prompt, yet it introduces abrupt visual changes and temporal discontinuities, as shown in Figure 3 (a). Conversely, retaining the entire KV cache often prevents the model from following new prompts, or adapting to new prompts after a delay, because the cache is saturated with information from the previous prompt, as shown in Figure 3 (b). Based on this observation, we first diagnose why prompt switching is hard for streaming video generators. In DiT (Peebles & Xie, 2023) architectures, cross-attention and self-attention layers alternate. During generation, large amounts of information from the previous prompt are repeatedly injected through cross-attention layers and then propagated forward by self-attention, so that this prompt signal is written into the running KV cache. Consequently, when the prompt is switched, the model still carries residual semantics of the old prompt in the cache. And in certain instances, this results in inconsistent adherence to the new prompt.

To address this issue, we introduce KV recache. At a prompt switch boundary, we recompute the KV cache using **the already generated frames** together with **the new prompt**, effectively erasing residual information from the previous prompt while keeping the motion and visual cues that guarantee temporal continuity. Concretely, at the first post-switch frame, we encode the generated

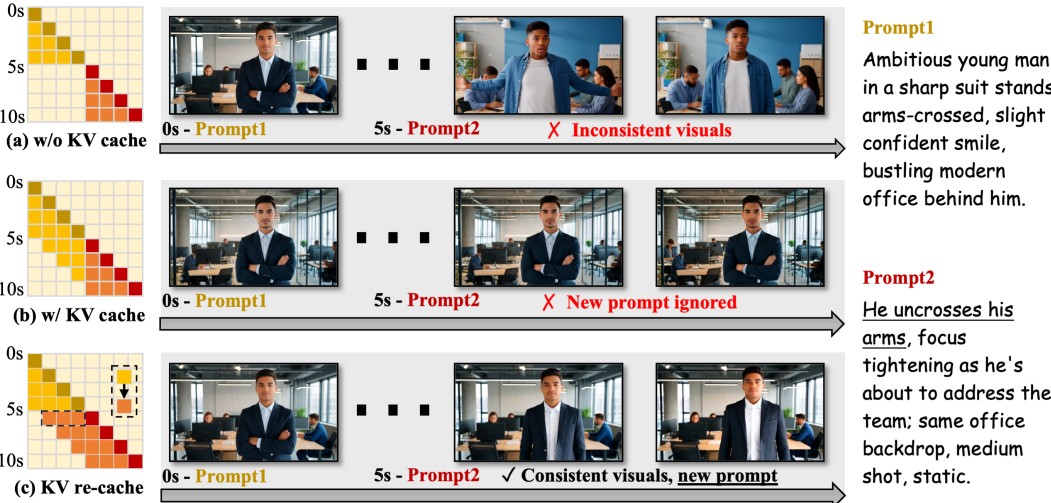

Figure 3: Prompt switching under different KV-cache strategies. **(a) w/o KV cache:** New prompt takes effect, but transitions are abrupt and visuals are inconsistent. **(b) w/ KV cache:** Smooth continuity, but the new prompt is not followed (lag or ignore). **(c) KV re-cache:** Smooth, visually consistent transitions with full new-prompt compliance.

video prefix as the visual context and pair it with the next prompt to rebuild the cache; subsequent steps then proceed normally using this refreshed cache. In this way, the cache retains the visual state of the ongoing video, but the prompt semantics now cleanly correspond to the active prompt, enabling improved semantic alignment without visual discontinuities.

To ensure train–inference alignment, we integrate the recaching operation into our training loop (Figure 4). When a training iteration contains a prompt switch, we (i) perform recache once, (ii) continue rollout with the updated cache, and (iii) in distillation, feed the teacher model with the new prompt as well, so the student is supervised under the exact post-switch condition it will face at inference. This training scheme further removes the train-inference mismatch. Models trained with recache therefore exhibit both strong temporal smoothness and fast semantic convergence to the next prompt at inference, as illustrated in Figure 3 (c). In terms of efficiency, recaching is invoked only once per training sample. The added cost is thus minimal; for a 10s video with a single switch, recaching introduces only about 6% extra time cost compared to no recaching usage.

Moreover, although training includes only one prompt switch per long sequence, this mechanism generalizes well during inference. The model supports interactive inference with multiple prompt switches by performing a single recaching step at each boundary. Given $n + 1$ prompts and $n$ switch points, the generator rolls out causally, applies KV recaching at each switch, and continues producing frames semantically aligned with the active prompt while maintaining smooth transitions. A detailed illustration of this procedure is outlined in Appendix Algorithm 2.

## 3.2 STREAMING LONG TUNING

LongLive builds upon causal frame-level AR video generators. These models are trained only on short clips. At inference, they produce long videos via a rolling, fixed-length context window that repeatedly feeds the model its own outputs. As the rollout continues, small prediction errors accumulate and the context inside the window becomes progressively noisier, so the model conditions on a more degraded self-generated history. Since such long-range, self-generated contexts were absent in training, this *train-short–test-long* regime induces content drift and breaks consistency over long horizons. To address this mismatch, we propose a *train-long–test-long* strategy. During training, the model synthesizes long sequences by conditioning on its own imperfect predictions, with supervision applied throughout the entire rollout. This exposes the model to extended, self-generated, and progressively degraded frames already in training, aligning training with inference, mitigating error accumulation to improve fidelity and consistency.

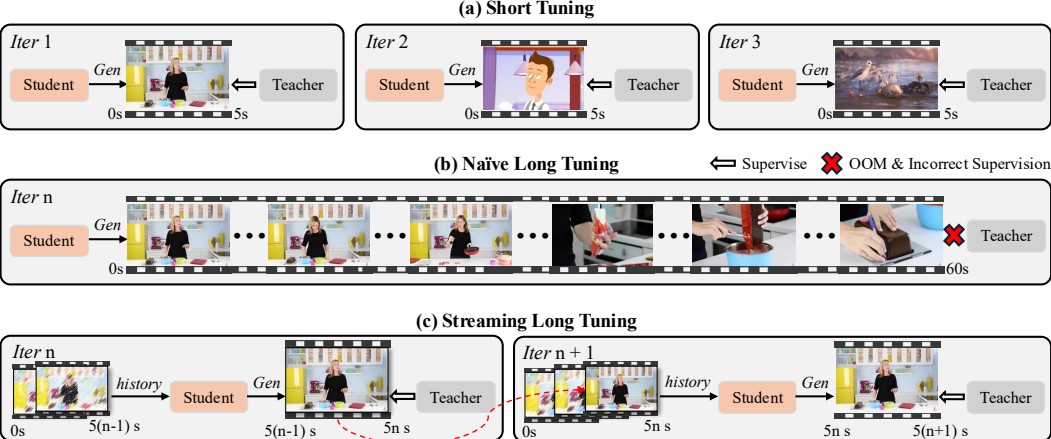

Figure 4: The streaming long tuning pipeline. (a) **Short tuning**: only 5s clips are supervised, like Self-Forcing (Huang et al., 2025), leading to quality loss on long videos. (b) **Naive long tuning**: naively scaling to long sequences causes incorrect teacher supervision and OOM. (c) **Streaming long tuning**: our approach trains on long sequences by reusing the historical KV cache each iteration to generate the next 5s clip, then supervising it with the teacher.

Self-supervision (Huang et al., 2025) methods are able to avoid collecting a large long-video dataset. It requires no real video data: a pretrained teacher provides synthetic supervision that guides the student to match the teacher's output distribution. However, two practical challenges arise in this method. First, the teacher itself is typically trained for short clips and thus cannot reliably supervise an entire long sequence end-to-end. Second, naïvely unrolling and backpropagating through long sequences easily triggers out-of-memory (OOM) issues and is computationally wasteful.

To address these two challenges, we introduce a streaming long tuning procedure (Figure 4) that learns on long videos while keeping memory and supervision local and reliable. In the first iteration, the generator samples a short video clip (e.g., 5s) from scratch, and we apply DMD (Yin et al., 2024b;a) on this short clip. In subsequent iterations, the generator extends the short clip from the previous iteration, and produce the next short clip conditioned on the previously stored KV cache, and we again apply DMD only to this newly generated clip. We repeat this rolling extension until the video reaches a preset maximum length, then fetch a new batch and restart from scratch. This schedule mirrors the inference-time rollout and thus reduces train–test inconsistency. At each iteration, the teacher provides reliable supervision for the current short clip (where it is competent), and the collection of per-clip supervisions provides global guidance for the full sequence. In practice, we detach the already generated frames so they act as a constant causal context. The gradients are computed only for the current generated clip. Consequently, memory usage is limited by the clip duration, avoiding OOM. A detailed illustration of this process appears in Appendix Algorighm 1.

Our study reveals that tuning on long videos is not only critical for the performance of long video generation, but also a prerequisite for efficient long inference strategies. These strategies include window attention and frame sink, which significantly improve inference speed.

### 3.3 EFFICIENT LONG INFERENCE

**Short-window Attention**  In long video generation, the cost of dense causal attention grows quadratically with the sequence length, making naive inference prohibitive on long videos. Motivated by evidence of temporal locality in video generation: nearby frames contribute more to predicting the next one (Gu et al., 2025; Zhang & Agrawala, 2025), we adopt local window attention during inference and during streaming tuning. Limiting attention to a fixed temporal window reduces both computation and memory. Attention complexity becomes proportional to the window size rather than the growing sequence length, and the KV cache needed per layer scales with the window rather than the total video. However, window size introduces a quality–efficiency trade-off. We generate 20-second videos using different attention window settings, as shown in the first and

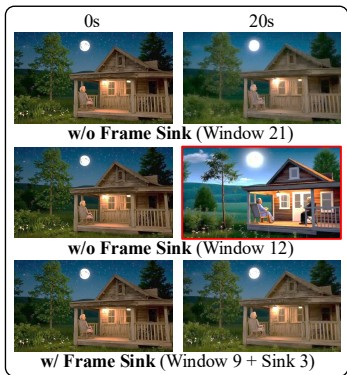 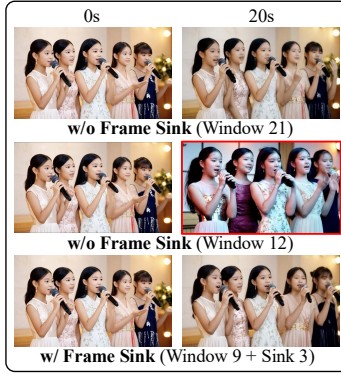 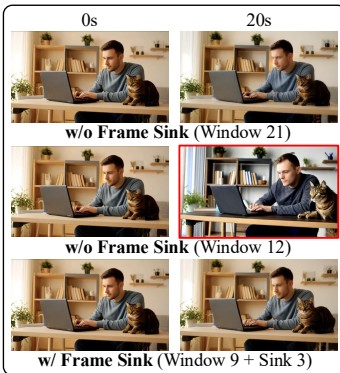

Figure 5: Comparison in a 20-second generated video of long window attention (Window 21 latent frames), short-window attention (Window 12), and short-window + frame-sink (Window 9 + Sink 3). Shorter windows boost efficiency but weaken long-range consistency; adding a frame-sink restores consistency while keeping the efficiency gains.

second rows in Figure 5. Larger windows retain more temporal context and yield stronger long-range consistency, but incur higher latency and memory. Shrinking the window improves efficiency at the cost of consistency, since distant but critical cues disappear from the receptive field.

**Frame Sink**    Prior work reported that attention-sink tokens alone do not prevent long-rollout collapse in video models (Huang et al., 2025). In contrast, we empirically find that attention sinks become effective once long-rollout collapse is addressed via streaming long tuning. Serving as persistent global anchors, attention sinks markedly improve long-range temporal consistency, thereby mitigating the quality–efficiency trade-off when using short-window attention. As shown in the third row of Figure 5, adding a frame-sink greatly boosts long-range consistency under a short window while maintaining low cost. Concretely, we fix the first frame chunk of the video as global sink tokens; these tokens are permanently retained in the KV cache and concatenated to every attention block's keys and values, making them globally attendable even with local-window attention. The remainder of the KV cache uses a short rolling window and is evicted normally. In experiments, a short-window with a frame-sink preserves high long-video quality while reducing end-to-end compute time by 28% and peak memory by 17% on a single H100 GPU.

**Consistency between Training and Inference**    We integrate short-window attention and the frame sink into streaming tuning to align train-test behavior and improve efficiency. Let the local attention window be $W$ frames and the supervised clip length (from the teacher) be $T$ frames. At each training step, we keep (i) the KV cache from the last $W$ frames of the preceding context *without gradients* and (ii) the full KV cache of $T$ frames for the current supervised clip *with gradients*. We also maintain $S$ sink tokens (the first two frames) that are never evicted and are concatenated to every layer's KV so they remain globally attendable. Consequently, the resident KV size per step is $O(W+T+S)$ and does *not* grow with total video length, preventing OOM on very long rollouts. The sinks stabilize identity and scene semantics, allowing us to train with the same shortened window used at inference. For KV re-caching, we rebuild the cache from only the most recent $W$ generated frames, which refreshes semantics while preserving local continuity and saves the re-caching cost.

## 4  EXPERIMENT

**Implementation**    We build LONGLIVE on Wan2.1-T2V-1.3B (Wan et al., 2025), which produces 5s clips at 16 FPS and $832 \times 480$ resolution. We first adapt the pretrained model into a few-step causal-attention model using a self-forcing (Huang et al., 2025) improved DMD pipeline (DMD + DMD2 few-step) on VidProM (Wang & Yang, 2024) data, while enabling our short-window attention and the frame sink (we keep all tokens from the first frame chunk as sink tokens). We then perform streaming long tuning on a 60s sequence that contains a single prompt switch. To construct this switch-prompt dataset, we prompt Qwen2-72B-Instruct (Yang et al., 2024a) to generate

Table 1: **Comparison with relevant baselines.** We compare LONGLIVE with representative open-source video generation models of similar parameter sizes and resolutions. Evaluation scores are calculated on the standard prompt suite of VBench (Huang et al., 2024a). FPS - a single H100 GPU.

| Model | #Params | Resolution | Throughput (FPS) ↑ | Evaluation scores ↑ | | |
|---|---|---|---|---|---|---|
| | | | | Total | Quality | Semantic |
| *Diffusion models* | | | | | | |
| LTX-Video (HaCohen et al., 2025) | 1.9B | 768×512 | 8.98 | 80.00 | 82.30 | 70.79 |
| Wan2.1 (Wan et al., 2025) | 1.3B | 832×480 | 0.78 | 84.26 | 85.30 | 80.09 |
| *Autoregressive models* | | | | | | |
| SkyReels-V2 (Chen et al., 2025a) | 1.3B | 960×540 | 0.49 | 82.67 | 84.70 | 74.53 |
| MAGI-1 (Teng et al., 2025) | 4.5B | 832×480 | 0.19 | 79.18 | 82.04 | 67.74 |
| CausVid (Yin et al., 2025) | 1.3B | 832×480 | 17.0 | 81.20 | 84.05 | 69.80 |
| NOVA (Deng et al., 2025) | 0.6B | 768×480 | 0.88 | 80.12 | 80.39 | 79.05 |
| Pyramid Flow (Jin et al., 2025) | 2B | 640×384 | 6.7 | 81.72 | 84.74 | 69.62 |
| Self Forcing, chunk-wise (Huang et al., 2025) | 1.3B | 832×480 | 17.0 | 84.31 | 85.07 | **81.28** |
| Self Forcing, frame-wise (Huang et al., 2025) | 1.3B | 832×480 | 8.9 | 84.26 | 85.25 | 80.30 |
| LONGLIVE | 1.3B | 832×480 | **20.7** | **84.34±0.52** | **85.72±1.07** | 79.62±2.73 |

Table 2: Interactive long video evaluation: Quality scores are reported on the whole 60s sequence. CLIP scores are reported on 10s video segments with the same semantics (↑ higher is better).

| Method | Quality Score ↑ | CLIP Score ↑ | | | | | |
|---|---|---|---|---|---|---|---|
| | | 0–10 s | 10–20 s | 20–30 s | 30–40 s | 40–50 s | 50–60 s |
| SkyReels-V2 (Chen et al., 2025a) | 79.85±0.92 | 21.34±1.98 | 23.15±1.85 | 23.54±2.21 | 17.95±1.46 | 20.12±0.97 | 19.25±0.97 |
| Self-Forcing (Huang et al., 2025) | 82.15±1.05 | 27.92±0.86 | 25.34±1.75 | 23.12±1.46 | 22.45±1.58 | 23.67±0.82 | 23.55±1.87 |
| LONGLIVE | **85.02±1.12** | **29.45±1.72** | **26.15±1.03** | **25.10±1.34** | **24.85±0.88** | **24.90±1.13** | **24.65±1.38** |

follow-up prompts conditioned on each original VidProM prompt. During training, each iteration continues the model's own rollout by generating the next 5s video clip until a maximum length of 60s is reached; each batch includes exactly one prompt switch with the switch time sampled uniformly from 5s to 55s. When a switch occurs, we apply KV-recache. During streaming long tuning, we also keep the same short-window attention and frame-sink settings. This training procedure takes about 12 hours on 64 H100 GPUs. Notably, LONGLIVE supports any model capable of autoregressive rollout with a KV cache. We implement LONGLIVE on a linear-attention AR model, SANA-Video (Chen et al., 2025b), achieving further acceleration on long-video generation.

## 4.1 SHORT VIDEO GENERATION

We first evaluate LONGLIVE's short-video generation on VBench using their official prompts, and compared it with relevant open-source video generation models of similar scale, including LTX-Video (HaCohen et al., 2025), Wan2.1 (Wan et al., 2025), SkyReels-V2 (Chen et al., 2025a), MAGI-1 (Teng et al., 2025), CausVid (Yin et al., 2025), NOVA (Deng et al., 2025), Pyramid Flow (Jin et al., 2025), and Self-forcing (Huang et al., 2025). All scores are normalized using the same numerical system with VBench. On 5-second clips, LONGLIVE matches the strongest baselines in total score, demonstrating excellent quality and stability, as shown in Table 1. Benefiting from the short window attention design, LONGLIVE is also the fastest among all the methods, reaching 20.7 FPS for real-time inference. It shows that LONGLIVE does not degrade the short-clip generation capability.

## 4.2 LONG VIDEO GENERATION

We evaluate LONGLIVE's single-prompt long-video generation on VBench-Long (Huang et al., 2024b) using its official prompt set. For each prompt, we generate a 30-second video and split it into clips according to the VBench-Long official scripts. We compare against three representative open-source models: SkyReels-V2 (Chen et al., 2025a), FramePack (Zhang & Agrawala, 2025), and Self-Forcing (Huang et al., 2025). Because FramePack is an I2V model, we first synthesize an initial frame from the same text prompt and feed it to FramePack; other T2V models generate directly from the prompt. We report the standard VBench-Long metrics for long-horizon quality and consistency in Table 3. LONGLIVE achieve the state-of-the-art performance, while being the fastest.

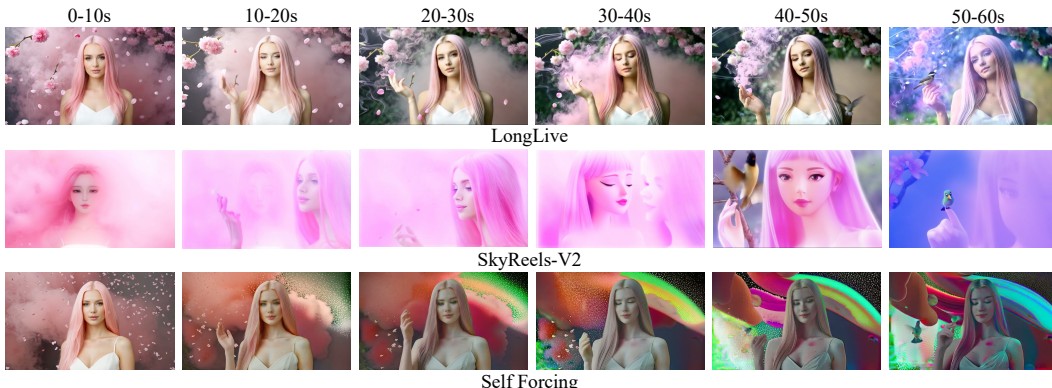

0-10s: Medium close-up of the serene model in a white gown amid drifting sakura petals and soft pink smoke.
10-20s: She slowly lifts her hand, fingertips grazing a petal/soft pink smoke
20-30s: A gentle breeze rises. The petals float softly in the air, creating a dreamy and ethereal atmosphere.
30-40s: She softly closes her eyes while holding the same poised gesture.
40-50s: Her fingers barely touching the delicate pink smoke, while a small bird flits in.
50-60s: the bird now perches on her outstretched finger; pink and cyan-blue smoke thickens and envelops the scene.

Figure 6: Qualitative comparison for interactive long video generation. LONGLIVE exhibits strong prompt compliance, smooth transitions, and high long-range consistency while sustaining high throughput. Compared to ours, SkyReels-V2 shows weaker long-range consistency, and Self-Forcing faces quality degradation on longer videos.

Table 3: Single-prompt 30s long video evaluation on VBench-Long.

| Model | Total Score ↑ | Quality Score ↑ | Semantic Score ↑ | Throughput (FPS) ↑ |
|---|---|---|---|---|
| SkyReels-V2 | 75.29 | 80.77 | 53.37 | 0.49 |
| FramePack | 81.95 | 83.61 | 75.32 | 0.92 |
| Self-Forcing | 81.59 | 83.82 | 72.70 | 17.0 |
| LONGLIVE | **83.52** | **85.44** | **75.82** | **20.7** |

Table 4: Ablation study on KV recache. KV recache achieves the best consistency score and CLIP score.

| Method | Background Consistency ↑ | Subject Consistency ↑ | CLIP Score ↑ | FPS ↑ |
|---|---|---|---|---|
| No KV cache | 92.75 | 89.59 | 28.95 | 22.8 |
| KV cache | 94.77 | 93.69 | 25.92 | 21.9 |
| KV recache | 94.81 | 94.04 | 27.87 | 20.7 |

## 4.3 INTERACTIVE LONG VIDEO GENERATION

For interactive long-form videos with multiple prompt switches, few existing methods support true streaming generation. We implemented this setting for two representative baselines: SkyReels-V2 and Self-Forcing. We then compare our approach against them. Because the standard VBench protocol is not directly applicable, we curated a custom set of 160 interactive 60-second videos, each comprising six successive 10-second prompts as the validation set. For long-horizon quality, we evaluate our 60s interactive videos on VBench-Long dimensions that support customized prompt videos, including subject_consistency, background_consistency, motion_smoothness, aesthetic_quality, and imaging_quality. For semantic adherence, we segment each video at prompt boundaries and compute clip-wise semantic score using CLIP (Radford et al., 2021) scores. Qualitative and quantitative results are shown in Figure 6 and Table 2, respectively. LONGLIVE exhibits strong prompt compliance, smooth transitions, and high long-range consistency while sustaining high throughput. In contrast, Self-Forcing degrades on longer horizons and, SkyReels-v2 shows weaker consistency. In terms of speed, LONGLIVE is more than 41× faster than SkyReels-v2 and slightly faster than Self-Forcing, even with KV re-cache, thanks to our short-window attention design. Please see our project page for more qualitative comparisons for interactive long video generation. Finally, a user study in which participants rated Overall Quality, Motion Quality, Instruction Following, and Visual Quality, *i.e.*, Figure 1 (right) further supports the effectiveness of our approach.

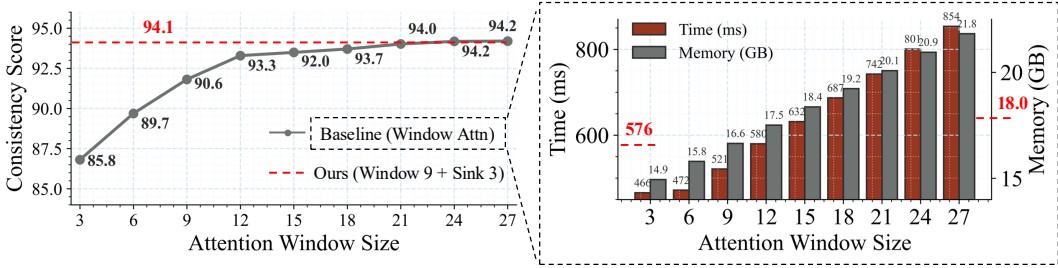

Figure 7: Ablation study on short window size and frame sink. Smaller windows reduce consistency, while enabling frame sink mitigates the drop.

### 4.4 KV Recache

In Table 4, we ablate KV caching strategies at prompt switches in a 10-second video setting with a single switch at the 5-second. We compare (i) No KV cache: clear the entire cache at the switch; (ii) KV-cache: retain the full cache unchanged; and (iii) KV-recache (ours): refresh the cache by recomputing key–value states conditioned on the preceding frames and the new prompt. We assess visual consistency with VBench Background Consistency and Subject Consistency, and measure semantic score with the CLIP model. Clearing the cache breaks long-range consistency, causing abrupt visual changes. Retaining the cache preserves continuity but induces prompt inertia: the model sticks to the previous prompt, yielding a lower semantic score on the switched prompt. Our KV recache maintains continuity while restoring compliance to the switched prompt. Please see Figure 3, Appendix Figure E, and the demo page for more qualitative comparisons on KV recache.

### 4.5 Short-Window Attention and Frame Sink

In Figure 7, we ablate short-window attention and the frame-sink under a 10-second generation setting. We vary the local-attention window from 3 to 27 latent frames, and additionally evaluate a configuration with 9 local latent frames plus 3 sink latent frames (effective window size 12). Long-range consistency is measured using VBench-Long (Huang et al., 2024b) (Background Consistency and Subject Consistency). Consistency improves as the attention window grows and saturates around a 24-frame window, revealing a clear quality–efficiency trade-off: larger windows retain more temporal context but increase latency and memory, while smaller windows are cheaper but less consistent. Our frame-sink mechanism mitigates this trade-off by recovering long-range context without attending to the full history: the 9-local + 3-sink setting achieves consistency close to a 21-frame window while preserving the speed and memory footprint of a short window.

## 5 Conclusion

In this work, we introduce LONGLIVE, a frame-level AR framework for real-time and interactive long video generation. To maintain visual smoothness and semantic adherence during prompt switches in interactive settings, we propose a KV-recache technique. We present a streaming long tuning strategy that enables direct training on long videos, ensuring high-quality outputs. We further introduce short window attention and frame sink to accelerate long video generation while preserving visual consistency. Experimental results demonstrate that LONGLIVE can efficiently fine-tune a model for long-video AR generation in only 32 GPU-days. Moreover, tuning on long videos is essential not only for long video generation but also as a prerequisite for efficient inference (e.g., window attention with frame attention sink), substantially improving inference speed. During inference, it achieves 20.7 FPS inference on a single NVIDIA H100 GPU, and supports up to 240-second video generation while maintaining high fidelity and temporal coherence. Using INT8 quantization, LONGLIVE compresses from 2.7 GB to 1.4 GB, with minimal performance degradation. LONGLIVE also supports INT8-quantized inference, incurring only marginal quality loss. We provide further results, analyses, implementation details, and qualitative showcases in the Appendix.

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

# APPENDIX

## A  ETHICS STATEMENT

This study uses a self-supervised, efficient fine-tuning procedure and does not introduce any additional external video datasets for training. All text prompts leveraged in self-supervised training, generated from Qwen2-72B-Instruct (Yang et al., 2024a), are clean, safe, and for academic research purposes only.

## B  REPRODUCIBILITY STATEMENT

To facilitate reproducibility, we will open-source this project, including both training and inference code as well as model weights. In addition, we provide the full training procedure and implementation details in Section 4 and Section F.

## C  USE OF LARGE LANGUAGE MODELS

During manuscript preparation, we used large language models—GPT-5 (OpenAI, 2025)—strictly for language polishing of paragraphs and sentences (grammar, flow, and tone). These tools were not used to generate ideas, design experiments, or determine conclusions. All technical content, methodology, and interpretations were written, verified, and approved by the authors. To reduce risks of factual drift or citation errors, we required human review of every model-edited sentence and cross-checked all references against primary sources. The authors take full responsibility for the accuracy and integrity of the manuscript.

## D  GENERAL RELATED WORK

### D.1  DIFFUSION-BASED LONG VIDEO GENERATION

Recent advances in diffusion models (Villegas et al., 2023; He et al., 2022; Chen et al., 2024b; Wang et al., 2025; Guo et al., 2025; Dalal et al., 2025) have explored long video generation. Phenaki (Villegas et al., 2023) compresses video into discrete tokens, enabling variable-length generation from open-domain text. NUWA-XL (Yin et al., 2023) extends diffusion to extremely long sequences via a coarse-to-fine "diffusion over diffusion" framework, generating global keyframes and filling intermediate frames in parallel. LVDM (He et al., 2022) leverages a compact 3D latent space with hierarchical generation. LaVie (Wang et al., 2025) proposes a cascaded pipeline, with joint fine-tuning, rotary position encoding, and temporal attention. SEINE (Chen et al., 2024b) employs smooth shot transitions using a stochastic masking-based diffusion model. LCT (Guo et al., 2025) expanded pre-trained short-video models to scene-level contexts for multi-shot coherence, via large-scale fine-tuning. Other approaches (Dalal et al., 2025) use a test-time training technique to generate minute-long videos. TECO (Yan et al., 2023), MALT Diffusion (Yu et al., 2025), and Rolling Diffusion Ruhe et al. (2024) all study long-horizon sequence generation or prediction by designing architectures and training schemes that better preserve temporal coherence over many frames (e.g., transformer-based predictive models in TECO on navigation-style video benchmarks, memory-augmented latent transformers in MALT on UCF-101 / Kinetics-600, and sliding-window diffusion with per-frame time reparameterization in Rolling Diffusion for Kinetics-600 video prediction and other temporal data). However, they are primarily evaluated on relatively small-scale video prediction or class-conditioned generation datasets such as UCF or Kinetics, are not designed as large-scale text-to-video systems, and do not target real-time streaming generation, which makes them conceptually related but orthogonal to the real-time, text-driven long-video setting addressed by LongLive. Although these models can generate long-duration videos, they often incur heavy computational costs, motivating more efficient and real-time solutions.

Several recent works extend the generation length of diffusion models in a training-free manner. RI-FLEx (Zhao et al., 2025) conducts video length extrapolation by adjusting the intrinsic frequency of position embeddings, mitigating temporal repetition and motion slowdown. FreeNoise (Qiu et al., 2024) uses a noise rescheduling strategy and window-based temporal attention. FreeLong (Lu et al.,

2024) blends temporal frequency components at inference. FreeLong++ (Lu & Yang, 2025) introduces multi-band spectral fusion to capture and fuse multi-frequency temporal information. In these training-free settings, models achieve at most a 4–8× extension in length (up to 40 seconds), which remains inadequate for long-form scenarios.

## D.2 AUTOREGRESSIVE LONG VIDEO GENERATION

A growing number of works (Chen et al., 2024a; Song et al., 2025; Mao et al., 2025; Yuan et al., 2025; Zhang & Agrawala, 2025; Gao et al., 2025; Henschel et al., 2025; Gao et al., 2025) integrate diffusion modeling with AR prediction, an intermediate paradigm between purely diffusion-based approaches and purely AR approaches. Diffusion-forcing (Chen et al., 2024a) formalizes this hybrid paradigm by injecting noise into future tokens and training the model to denoise them, combining diffusion quality with AR efficiency. StreamingT2V (Henschel et al., 2025) extends this idea with short and long-term memory modules for coherent text-to-video generation. Pyramidal-flow (Jin et al., 2025) proposes a multi-scale flow matching design to reduce computation. History-guided video diffusion (Song et al., 2025) further incorporates flexible-length historical context to improve temporal consistency over extended rollouts. SkyReels-V2 (Chen et al., 2025a) couples diffusion forcing with a film-structure planner and multimodal controls. FramePack (Zhang & Agrawala, 2025) compresses input frames into a fixed-size context to address memory and efficiency bottlenecks. Lumos-1 (Yuan et al., 2025) employs large language models (LLMs) style architectures, integrating spatiotemporal modeling under the diffusion-forcing framework. Most recently, LongVie (Gao et al., 2025) introduces multimodal-guided control, unified noise initialization, and degradation-aware training. Recent efforts (Yin et al., 2025; Huang et al., 2025; Gu et al., 2025; Teng et al., 2025; Zhou et al., 2025; Deng et al., 2025) have advanced causal AR-based models for long video generation. CausVid (Yin et al., 2025) reformulates bidirectional video diffusion into a causal AR process, using distribution matching distillation to compress multi-step denoising into a few steps. FAR (Gu et al., 2025) further enhances AR generation by combining a high-resolution short-term context with a compressed long-term context via flexible positional encoding. MAGI-1 (Teng et al., 2025) scales AR video generation to large models and datasets through chunk-wise prediction. Most recently, Self-forcing (Huang et al., 2025) addresses the train–test gap in AR video diffusion by simulating inference conditions during training, rolling out generation with KV cache, and conditioning on model outputs. Despite the promise of purely AR for long text-to-video generation, achieving real-time efficiency and maintaining high quality simultaneously remains an open challenge.

Recent works have begun exploring interactive video generation, where users can directly influence generation in real time through text or keyboard prompts. The Matrix (Feng et al., 2024) demonstrates infinite-horizon world generation with first- and third-person control, using a shifted window denoising process. Yume (Mao et al., 2025) builds an interactive world generation pipeline capable of constructing explorable environments from a single image, video, or text, allowing responsive user navigation. Matrix-Game (Zhang et al., 2025) employs large-scale pretraining and action-labeled finetuning to produce controllable, high-fidelity video conditioned on reference frames, motion context, and user actions. While effective, these methods are specifically tailored for interactive video generation in video game environments, such as Minecraft and GTA. MAGI-1 (Teng et al., 2025) supports general interaction, but its prompt switching requires manual adjustment of KV-cache windows at different steps, which complicates practical use.

## E TRAINING PROMPT GENERATION

LONGLIVE does not require video data since we adopt a self-training method. It relies only on a set of prompts to teach the model with interaction ability (Yang et al., 2024b; 2021; 2023). To efficiently produce appropriate, reasonable, and safe interactive prompts, we employ the Qwen2-72B-Instruct (Yang et al., 2024a) LLM. Given a source prompt from `VidProM` (Wang & Yang, 2024), we instruct Qwen2-72B-Instruct to synthesize the next scene under several constraints. The instruction template is shown below.

```
You are a video-prompt generation specialist. Your task
• Receive an ORIGINAL_PROMPT for the first part of a continuous shot.
• Write one stand-alone English paragraph (80{100 words) that shows the
```

**Algorithm 1** Streaming Long Tuning

**Require:** Causal video generator $G_\theta$, Prompt set $\mathcal{P}$
**Require:** Video length $l_{\text{video}}$, Per clip length $l_{clip}$
1: **while** not converged **do**
2:      Initialize KV cache $C \leftarrow []$
3:      Initialize current video length $l \leftarrow 0$
4:      Sample $(p,\ p_{\text{next}}) \sim \mathcal{P}$
5:      Sample switch index $s$
        $s \in \{1, 2, \ldots, \lfloor l_{\text{video}}/l_{\text{clip}} \rfloor - 1\}$
6:      $s \leftarrow s \cdot l_{\text{clip}}$
7:      **if** $l \geq l_{\text{video}}$ **then**
8:         $C \leftarrow []; \quad l \leftarrow 0$
9:         Resample $(p,\ p_{\text{next}})$ and $s$
10:     **end if**
11:     $p_{\text{active}} \leftarrow \begin{cases} p, & \text{if } l < s \\ p_{\text{next}}, & \text{otherwise} \end{cases}$
12:     **if** $l = s$ **then**
13:        $C \leftarrow \texttt{recache}(G_\theta,\ \mathbf{v},\ C,\ p_{\text{active}})$
14:     **end if**
15:     $\mathbf{x} \leftarrow \texttt{generate\_next\_clip}(G_\theta,\ C,\ p_{\text{active}})$
16:     $\mathcal{L} \leftarrow \texttt{DMD\_Loss}(G_\theta,\ \mathbf{x},\ p_{\text{active}})$
17:     $\mathcal{L}.\texttt{backward}()$
18:     update generator parameter $\theta$
19:     $l \leftarrow l + l_{clip}$
20: **end while**

**Algorithm 2** Interactive Inference

**Require:** Causal video generator $G_\theta$
**Require:** Prompt sequence $\mathcal{P} = [p_0, \ldots, p_n]$, switch-index sequence $\mathcal{S} = [s_1, \ldots, s_n]$
**Require:** Number of video frames $N$, diffusion steps per frame $T$
1: Initialize model output $\mathbf{x} \leftarrow []$
2: Initialize KV cache $C \leftarrow []$
3: $p_{\text{active}} \leftarrow \mathcal{P}.\texttt{pop(0)}$
4: **for** $i = 1, \ldots, N$ **do**
5:     **if** $i \in \mathcal{S}$ **then**
6:        $p_{\text{active}} \leftarrow \mathcal{P}.\texttt{pop(0)}$
7:        $C \leftarrow \texttt{recache}(G_\theta,\ \mathbf{x},\ C,\ p_{\text{active}})$
8:     **end if**
9:     Initialize $x_{t_T}^i \sim \mathcal{N}(0, I)$
10:    **for** $j = T, \ldots, 1$ **do**
11:       Set $\hat{x}_0^i \leftarrow G_\theta(x_{t_j}^i; t_j, C,\ p_{\text{active}})$
12:       **if** $j = 1$ **then**
13:         $\mathbf{x}.\texttt{append}(\hat{x}_0^i)$
14:         $C \leftarrow G_\theta^{KV}(x_j^i,\ 0,\ C,\ p_{\text{active}})$
15:       **else**
16:         Sample $\epsilon \sim \mathcal{N}(0, I)$
17:         Set $x_{t_{j-1}}^i \leftarrow \Psi(\hat{x}_0^i, \epsilon, t_{j-1})$
18:       **end if**
19:     **end for**
20: **end for**
21: **return** $\mathbf{x}$

```
next moment of the same shot.
• **Add exactly one new action/object for the existing main subject.**
• Keep setting, subject, mood, style, camera scale, and camera movement
or angle exactly as in the ORIGINAL_PROMPT.
• Elements may vanish only if naturally obscured by the new action.
• Do **not** use phrases like *still, as before, continues* that reveal
you read the prior text.
• Use clear mid-level English; avoid rare or literary words.
• End the paragraph with **the same camera keywords that appear at the
end of the ORIGINAL_PROMPT**, separated by single spaces, no brackets.
• **Output format MUST be exactly one line, wrapped between <OUTPUT> and
</OUTPUT>.**
• Do **NOT** add explanations, greetings, headings, numbering, markdown,
or extra lines.
• Anything written outside the two tags will be ignored.
```

# F   TRAINING DETAILS

## F.1   IMPLEMENTATION

We first adapt the pretrained `Wan2.1-T2V-1.3B` into a chunk-wise autoregressive (AR) causal-attention model. First, we conduct an ODE initialization as the same as self-forcing. Then we train the model with DMD, but switch to short-window attention with frame-sink tokens: the chunk size is **3** latent frames, the local attention window is **9** frames, and the first chunk (3 latent frames) serves as the sink. After this initialization, we perform streaming long-tuning strictly following Algorithm 1: at each iteration, we roll out a $5\,\text{s}$ clip and supervise the student using `Wan2.1-T2V-14B` as the teacher. Optimization uses AdamW for both actor and critic with learning rates $\text{lr} = 1.0 \times 10^{-5}$ (actor) and $\text{lr}_{\text{critic}} = 2.0 \times 10^{-6}$; we set $\beta_1 = 0.0$, $\beta_2 = 0.999$ for the actor and $\beta_{1,\text{critic}} = 0.0$, $\beta_{2,\text{critic}} = 0.999$ for the critic. Training is conducted on $64$ GPUs with one sample per GPU (global batch size $= 64$). We apply EMA to the actor with decay 0.99, starting at step 200. The maximum

Table A: LoRA budget vs. performance on VBench-Long. A moderate budget approaches full-model quality with far fewer trainable parameters.

| LoRA rank | 32 | 64 | 128 | 256 | 512 | Full Model |
|---|---|---|---|---|---|---|
| Trainable Parameters | 44 M | 87 M | 175 M | 350 M | 700 M | 1.3 B |
| Total Score | 81.08 | 82.68 | 82.98 | **83.12** | 83.04 | 83.52 |

Table B: INT8-Quantized results on VBench. FPS is measured on a single NVIDIA 5090 GPU.

| Precision | Model Size | Throughput (FPS) | Total | Quality | Semantic |
|---|---|---|---|---|---|
| INT8 | 1.4 GB | 16.4 | 84.31 | 86.20 | 76.74 |
| BF16 | 2.7 GB | 12.6 | 84.87 | 86.97 | 76.47 |

sequence length is set to the target inference horizon; both $60\,\mathrm{s}$ and $240\,\mathrm{s}$ work well in practice. For the $60\,\mathrm{s}$ setting, we train for 3,000 iterations.

## F.2 LoRA Tuning

Motivated by LongLora (Chen et al., 2024c), we assume that improving the quality of long context does not require a full model fine-tuning. We therefore adopt LoRA tuning throughout the streaming long tuning procedure. Interestingly, we find that effective long-range generation demands relatively high adapter ranks; in our setup, the resulting adapters require 256 ranks, making roughly 27% of the model's parameters trainable. Even so, LoRA substantially reduces the training footprint, cutting the parameter/optimizer state to about 27% of that required by full fine-tuning (i.e., 73% savings).

We ablate LoRA tuning in Table A. We measure the 30s long-video quality by VBench-long. Scaling the LoRA budget improves quality until a saturation point, with the rank 256 configuration achieving the best while still training far fewer parameters than full fine-tuning.

## G Ablation Study on the Number of Sink Frames

We have compared no sink versus different numbers of initial sink frames. As shown in the Table C and Figure A below, using a small number of sink frames (S3W9) slightly improves VBench scores, while increasing the number of sink frames (S6W6, S9W3) makes the sink frame visually dominant and causes flashback-like, less coherent long-range dynamics. We will report the VBench numbers as a table and provide qualitative comparisons in the PDF appendix.

We have expanded the ablation on the frame-sink configuration, where we use the notation $\mathrm{S}x\mathrm{W}y$ to denote using $x$ sink frames followed by a sliding window of $y$ frames (e.g., S3W9 means 3 sink frames + a 9-frame window). As shown in Table C, introducing a small number of sink frames (S3W9) slightly improves VBench scores over the no-sink baseline (S0W12). However, when we further increase the number of sink frames (S6W6, S9W3), the sink frames become visually dominant and the VBench scores degrade, with the S9W3 setting exhibiting a clear collapse in both overall and consistency scores. The qualitative comparisons in Figure A corroborate this trend: for S0W12, temporal consistency is poor and the red boxes highlight noticeable late-stage drifting; S3W9 yields more stable and coherent dynamics; while S6W6 and S9W3 introduce sudden content changes and flickering frames, with the red boxes marking these abrupt transitions and artifacts.

## H Quantization

We quantize LONGLIVE to INT8 via post-training quantization (Li* et al., 2025). As shown in Table B, this reduces LONGLIVE's model size by $1.9\times$ and improves throughput by $1.3\times$, with minimal degradation on VBench (Table B).

Table C: Ablation on the frame-sink configuration in terms of VBench. We use the notation $SxWy$ to denote $x$ sink frames followed by a $y$-frame window. A small number of sink frames (S3W9) slightly improves the VBench score over the no-sink baseline (S0W12), while increasing the number of sink frames (S6W6, S9W3) degrades performance, with S9W3 exhibiting a clear collapse in the consistency score.

| Metric | S0W12 | S3W9 | S6W6 | S9W3 |
|---|---|---|---|---|
| VBench Score ↑ | 83.78 | 83.98 | 79.23 | 74.43 |
| Consistency Score ↑ | 93.30 | 94.10 | 94.22 | 54.65 |

Figure A: Qualitative comparison of different frame-sink configurations using the $SxWy$ notation (S: number of sink frames; W: number of window frames). For S0W12, temporal consistency is poor and the red boxes highlight noticeable late-stage drifting; S3W9 yields more stable, coherent dynamics; while S6W6 and S9W3 produce visually dominant sink frames with sudden content changes and flickering artifacts, as indicated by the red boxes.

## I INTERACTIVE LONG VIDEO SHOWCASES

We present interactive 60s videos generated with six sequential prompts in Figure B and Figure C. See our demo page in the supplementary materials for more examples.

## J LONG VIDEO SHOWCASES

We present single-prompt 60 s videos in Figure D. See our demo page in the supplementary materials for more examples.

## K KV RE-CACHING COMPARISON

We present qualitative results from the ablation study of KV re-caching in Figure E. See our demo page in the supplementary materials for more examples. No KV cache: New-prompt adherence but abrupt transitions and visual discontinuity. KV cache: Smooth visuals but new-prompt non-adherence (delayed or ignored). KV recache: Visual consistency and new-prompt adherence.

## L ULTRA-LONG VIDEO ABLITIES

LONGLIVE can train and test on ultra-long sequences. We conduct an experiment on a 240-second sequence, and it generates this ultra-long video smoothly and consistently. See our demo page in the supplementary materials for ultra-long examples.

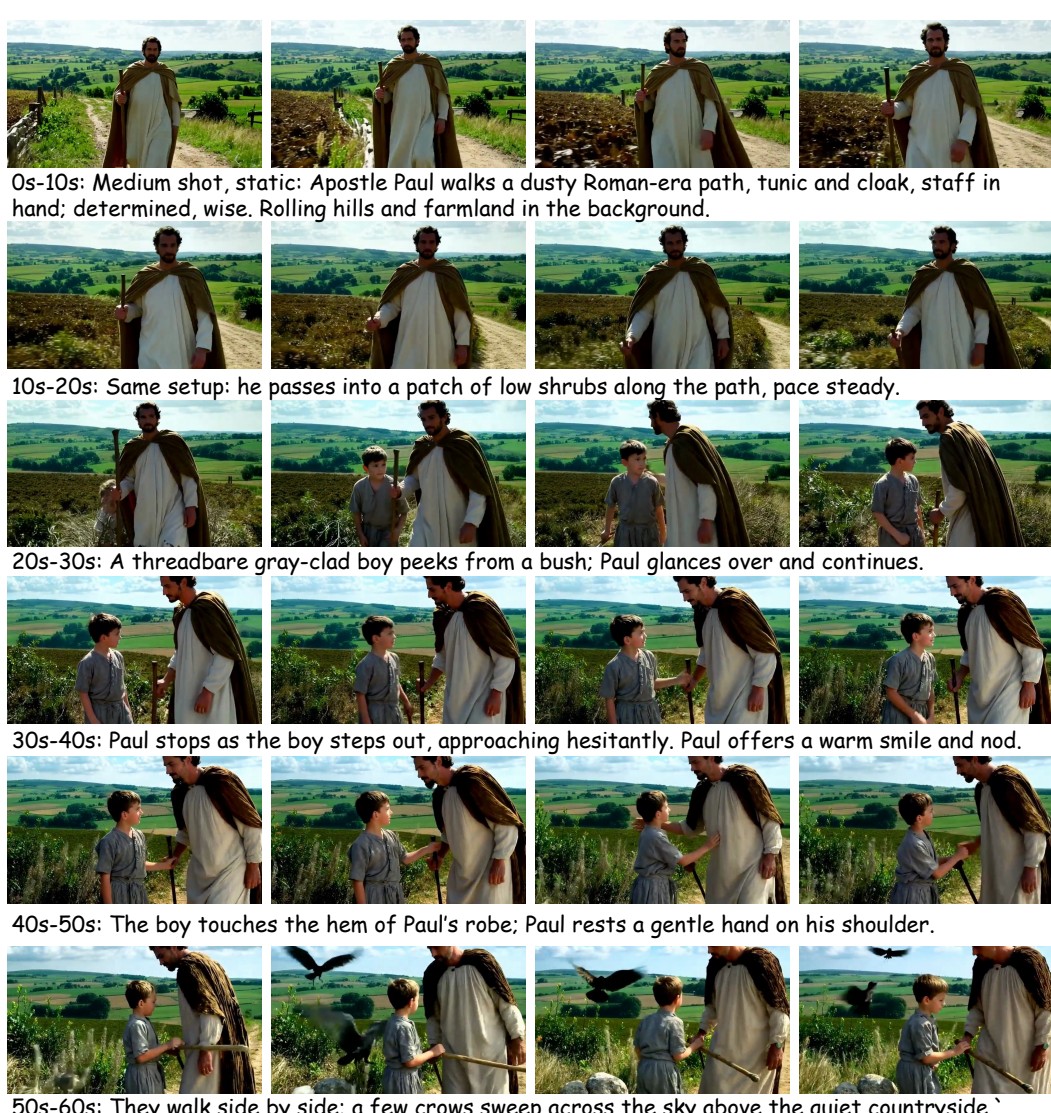

0s-10s: Medium shot, static: Apostle Paul walks a dusty Roman-era path, tunic and cloak, staff in hand; determined, wise. Rolling hills and farmland in the background.

10s-20s: Same setup: he passes into a patch of low shrubs along the path, pace steady.

20s-30s: A threadbare gray-clad boy peeks from a bush; Paul glances over and continues.

30s-40s: Paul stops as the boy steps out, approaching hesitantly. Paul offers a warm smile and nod.

40s-50s: The boy touches the hem of Paul's robe; Paul rests a gentle hand on his shoulder.

50s-60s: They walk side by side; a few crows sweep across the sky above the quiet countryside.`,

Figure B: Interactive 60s videos with sequential prompts. See our demo page in the supplementary materials for more examples.

## M    USER STUDY DETAILS

We conducted a user study to evaluate video quality across 48 questions spanning four dimensions: **Overall** (overall preference considering all factors), **Motion Quality** (smoothness/naturalness of motion; absence of jitter or discontinuity), **Instruction Following** (faithfulness to the given instruction/prompt), and **Visual Quality** (clarity, level of detail, and overall aesthetic quality). For each question, participants were shown a pair of videos together with the corresponding prompt and asked to choose **Model A**, **Model B**, or **Same** (no perceptible difference). The survey was distributed to 30 participants; we received **26** valid responses, yielding **1,248** total judgments ($26 \times 48$). Participants were instructed to watch both videos carefully and replay if needed before making a choice.

## N    LIMITATION ANALYSIS

LONGLIVE is an efficient fine-tuning scheme built on top of a pretrained base model, so its ultimate performance is bounded by the capacity and quality of that base model. In particular, we adopt a self-supervised fine-tuning strategy without additional curated real-video data. While this improves

efficiency and scalability, it also limits the method's ability to correct systematic errors or biases inherited from the base model. Consequently, the quality of any short segment (e.g., per 10-s clip) is unlikely to consistently exceed that of the base model, even if long-horizon consistency or instruction adherence improves. Therefore, our gains are primarily in adaptation and stabilization rather than absolute ceiling quality. Future work could incorporate supervised data to avoid the quality bound.

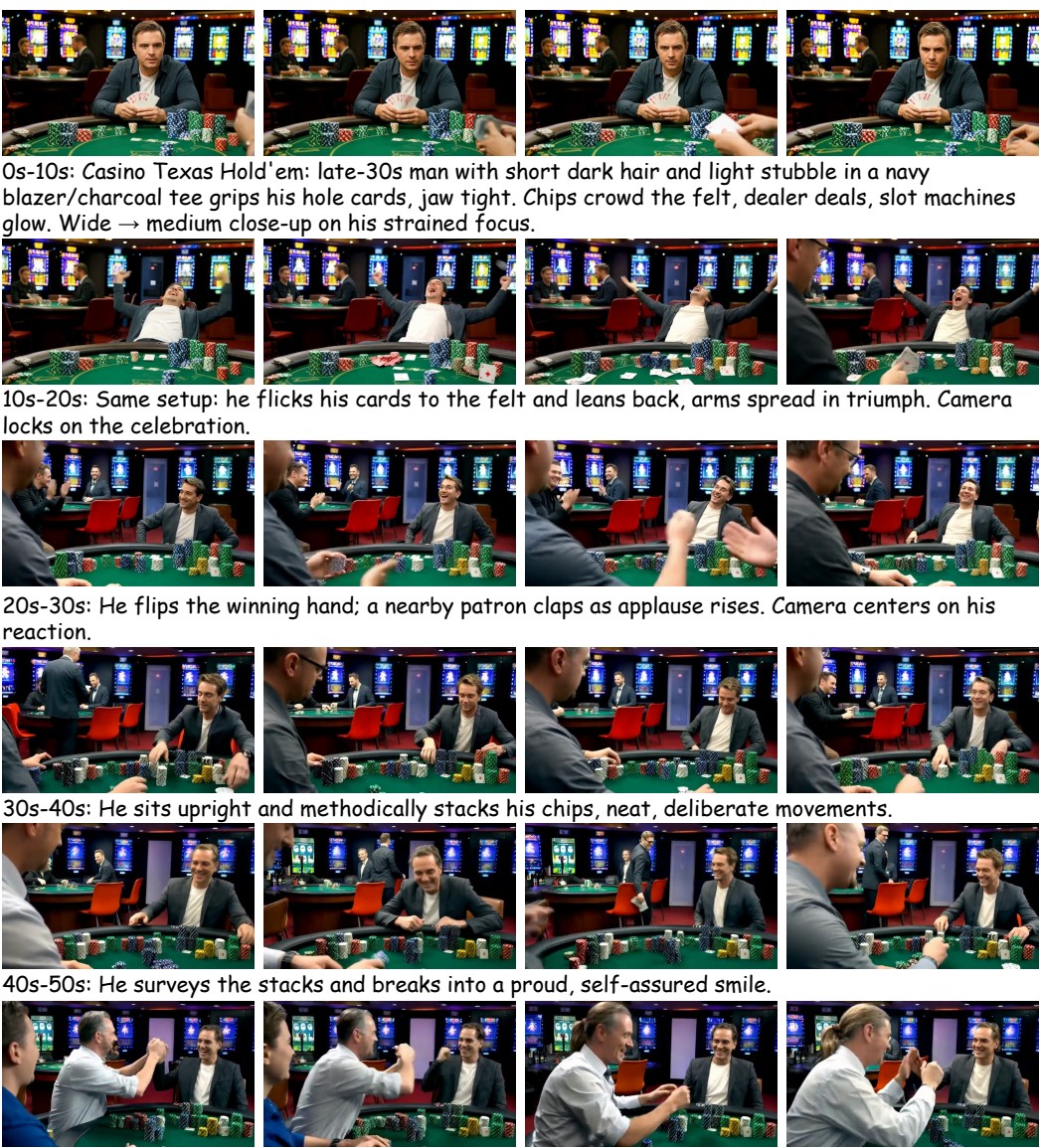

0s-10s: Casino Texas Hold'em: late-30s man with short dark hair and light stubble in a navy blazer/charcoal tee grips his hole cards, jaw tight. Chips crowd the felt, dealer deals, slot machines glow. Wide → medium close-up on his strained focus.

10s-20s: Same setup: he flicks his cards to the felt and leans back, arms spread in triumph. Camera locks on the celebration.

20s-30s: He flips the winning hand; a nearby patron claps as applause rises. Camera centers on his reaction.

30s-40s: He sits upright and methodically stacks his chips, neat, deliberate movements.

40s-50s: He surveys the stacks and breaks into a proud, self-assured smile.

50s-60s: He high-fives a patron; laughter and cheers ripple around the table. Same framing.

Figure C: Interactive 60s videos with sequential prompts. See our demo page in the supplementary materials for more examples.

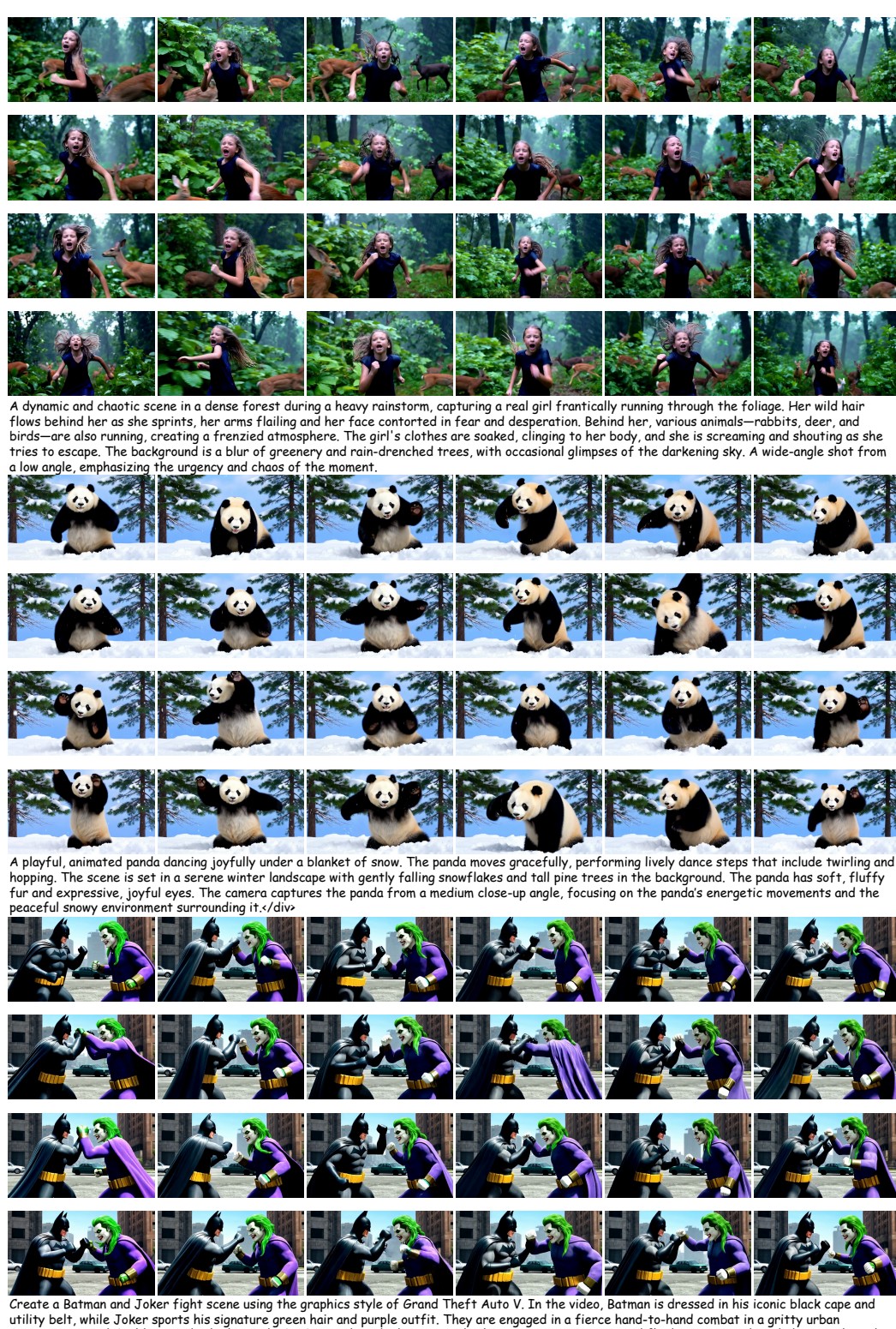

A dynamic and chaotic scene in a dense forest during a heavy rainstorm, capturing a real girl frantically running through the foliage. Her wild hair flows behind her as she sprints, her arms flailing and her face contorted in fear and desperation. Behind her, various animals—rabbits, deer, and birds—are also running, creating a frenzied atmosphere. The girl's clothes are soaked, clinging to her body, and she is screaming and shouting as she tries to escape. The background is a blur of greenery and rain-drenched trees, with occasional glimpses of the darkening sky. A wide-angle shot from a low angle, emphasizing the urgency and chaos of the moment.

A playful, animated panda dancing joyfully under a blanket of snow. The panda moves gracefully, performing lively dance steps that include twirling and hopping. The scene is set in a serene winter landscape with gently falling snowflakes and tall pine trees in the background. The panda has soft, fluffy fur and expressive, joyful eyes. The camera captures the panda from a medium close-up angle, focusing on the panda's energetic movements and the peaceful snowy environment surrounding it.

Create a Batman and Joker fight scene using the graphics style of Grand Theft Auto V. In the video, Batman is dressed in his iconic black cape and utility belt, while Joker sports his signature green hair and purple outfit. They are engaged in a fierce hand-to-hand combat in a gritty urban environment, with buildings and vehicles in the background. Both characters display intense expressions and fluid motions as they dodge, punch, and kick each other. The scene is captured from a dynamic third-person perspective, with occasional close-ups emphasizing their intense facial expressions and body language

Figure D: Single-prompt 60 s videos. See our demo page in the supplementary materials for more examples.

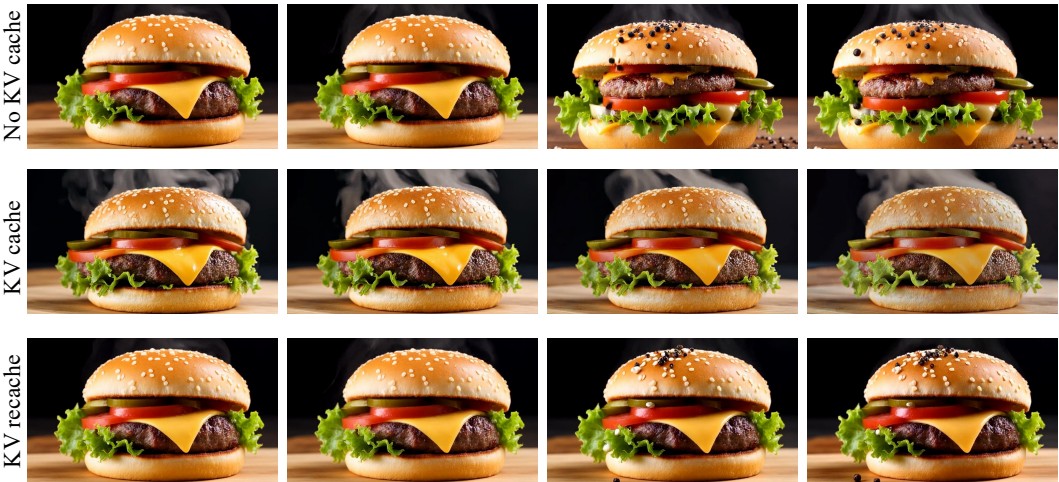

0s–5s: a steaming burger—seared patty (crisp edges, pink center), melted cheddar, lettuce, tomato, pickles, special sauce—on a lightly charred sesame bun.
5s–10s: fresh pepper sprinkles onto a hot patty under melted cheddar with lettuce, tomato, pickles, special sauce on a charred sesame bun.

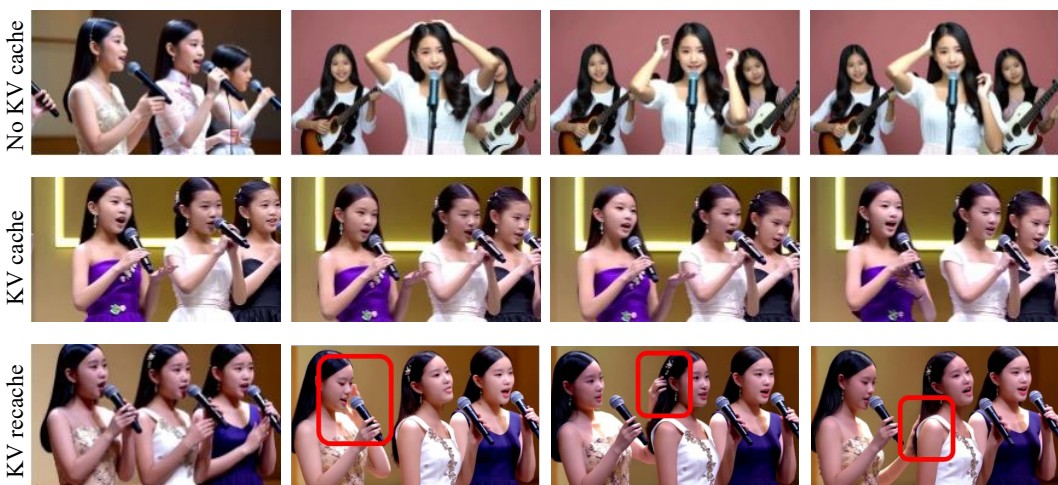

0s–5s: Young and beautiful girls singing...    5s–10s: One girl reaches up to adjust hair...

Figure E: We present qualitative results from the ablation study of KV re-caching. See our demo page in the supplementary materials for more examples. No KV cache: New-prompt adherence but abrupt transitions and visual discontinuity. KV cache: Smooth visuals but new-prompt non-adherence (delayed or ignored). KV recache: Visual consistency and new-prompt adherence.

