# OpenReview forum: "LongLive: Real-time Interactive Long Video Generation"
_ICLR.cc/2026/Conference — ICLR 2026 Poster_

### Official Review · Reviewer_1kFs · 2025-10-23

**Soundness:** 4
**Presentation:** 4
**Contribution:** 4
**Rating:** 8
**Confidence:** 3

**Summary:**

This paper investigates real-time, interactive minute-level video generation and frames it as a causal, frame-level autoregressive task. It introduces three synergistic ideas—KV-recache for smooth prompt switching, streaming long tuning to close the train-short/test-long gap, and short-window attention plus frame-sink tokens for accelerating inference. Extensive experiments on VBench and a new 160 interactive suite show that a 1.3 B model fine-tuned in only 32 GPU-days reaches 20.7 FPS on one H100 and outperforms diffusion and AR baselines in both short and  long clip settings. The manuscript is clearly structured and well illustrated.

**Strengths:**

- This paper tackles an important practical problem—interactive long video generation—and provides the first system that delivers >20 FPS at high resolution while keeping visual consistency.

- The authors propose KV-recache, a simple yet effective cache refresh that unsticks old-prompt semantics without breaking temporal continuity, nicely verified by ablation.

- Experiments are thorough: short vs. long, single vs. multi-prompt, throughput vs. quality, and ablations.

- The writing is accessible, with intuitive figures that contrast the proposed strategies and a reproducibility statement promising full code and weights.

**Weaknesses:**

Overall, the paper offers solid technical novelty and the core ideas are interesting; nevertheless, I have a few minor concerns and questions listed below.

- Why does SkyReels-V2 show a sudden performance jump in the 20–30 s interval? This spike suggests that the evaluation may be unstable. If so, I recommend re-running the evaluation with several random seeds (e.g., three) and reporting the mean and standard deviation to ensure reliability.

- The authors fine-tune on top of Wan2.1, yet the resulting model’s semantic-alignment score drops markedly, as shown in Table 1; no explanation for this degradation is provided. I encourage the authors to investigate and clarify the cause—either through analysis or additional ablation experiments—so that the community can better understand the trade-offs and further improve the model.

**Questions:**

See weakness

---

> ### Author Response · Authors · 2025-11-21
>
> Thank you for these valuable comments. We address your concerns point by point below.
> - W1: The evaluation may be unstable. SkyReels-V2 show a sudden performance jump in the 20–30 s interval.
>     - We appreciate the reviewer’s careful observations. For the SkyReels-V2 60 s evaluation, the sudden performance jump in the 20–30 s interval is indeed due to instability in our original evaluation. To address this, we re-ran Table 2 experiments under a more robust setting: for each test prompt, we now generate three videos with different random seeds and report the mean and standard deviation of the VBench scores and CLIP scores. With this updated protocol, the SkyReels-V2 curve becomes much smoother in the 20–30 s interval.
>
>
>         | Method | Quality Score ↑ | 0–10 s | 10–20 s | 20–30 s | 30–40 s | 40–50 s | 50–60 s |
>         | --- | --- | --- | --- | --- | --- | --- | --- |
>         | SkyReels-V2 | 79.85±0.92 | 21.34±1.98 | 23.15±1.85 | 23.54±2.21 | 17.95±1.46 | 20.12±0.97 | 19.25±0.97 |
>         | Self-Forcing | 82.15±1.05 | 27.92±0.86 | 25.34±1.75 | 23.12±1.46 | 22.45±1.58 | 23.67±0.82 | 23.55±1.87 |
>         | **LongLive** | **85.02±1.12** | **29.45±1.72** | **26.15±1.03** | **25.10±1.34** | **24.85±0.88** | **24.90±1.13** | **24.65±1.38** |
> - W2: Model’s short video semantic-alignment score drops markedly.
>     - The drop in the short-video semantic-alignment score in Table 1 is caused by the same evaluation instability. The semantic score itself has a larger deviation. To address this, we re-ran the 5 s short-video evaluation using three random seeds per test prompt and now report the mean and standard deviation of the VBench scores. In the updated Table 1, the semantic score differences are smaller and more consistent, leading to a more reliable assessment of our method’s short-video performance.
>         - Table1
>
>
>             | **Model** | **#Params** | **Resolution** | **Throughput (FPS) ↑** | **Total** | **Quality** | **Semantic** |
>             | --- | --- | --- | --- | --- | --- | --- |
>             | **LongLive** | 1.3B | 832×480 | 20.7 | 84.34 ± 0.52 | 85.72  ± 1.07 | 79.62  ± 2.73 |

---

> > ### Comment · Reviewer_1kFs · 2025-11-22
> >
> > Thank you for the detailed and thoughtful rebuttal.
> >
> > I appreciate the clarifications and the additional experiments provided. The authors have adequately addressed all of my concerns, and I have no further questions at this time.

---

### Official Review · Reviewer_AEHT · 2025-10-30

**Soundness:** 3
**Presentation:** 3
**Contribution:** 3
**Rating:** 6
**Confidence:** 4

**Summary:**

This paper presents LongLive, a method for fine-tuning pretrained video diffusion models for long video generation. First, it introduces KV-Recache, which regenerates the KV cache using the previously generated frames together with the new prompt when a prompt change occurs. This allows the model to generate videos that remain visually coherent with past frames while adapting better to the new prompt, rather than being stuck with the previous inputs. Moreover, the authors progressively extend the generation length by fine-tuning DMDs with an increasing number of video clips, where the gradient is computed only over the last video segment. The paper also proposes several additional techniques, such as applying an attention sink by maintaining the first frame as a global anchor. Leveraging all these techniques, LongLive demonstrates real-time long video generation (up to 240 seconds) on a single H100 GPU with minimal quality degradation.

**Strengths:**

- The paper is generally well-written and easy to follow.
- Real-time long video generation using pretrained diffusion models is a very challenging but important problem, and I think this paper provides quite effective techniques to solve this.
- For me, several techniques are simple (which is good) yet novel and effective, such as KV recache.
- I really like the supplementary material that the authors attached.

**Weaknesses:**

- There are several missing works in the literature on long video generation, such as AAPT [1], TECO [2], MALT [3], and Rolling Diffusion [4]. I recommend that the authors include a discussion of these and other relevant works that have not been covered in the paper.
- I also believe that having a long context is another important aspect in developing long video generation models. While being efficient, the proposed model inherently has a relatively short context length (mainly due to the attention sink that focuses on the first frame), which limits its effective context window. The method is indeed effective at generating visually plausible long videos in real time; however, it might struggle in scenarios that require longer temporal dependencies — for example, when a person disappears and reappears in the same scene, or when generating long videos with multiple scene transitions.

[1] Lin et al., Autoregressive Adversarial Post-Training for Real-Time Interactive Video Generation, 2025.
[2] Yan et al., Temporally Consistent Transformers for Video Generation, ICML 2023.
[3] Yu et al., MALT Diffusion: Memory-Augmented Latent Transformers for Any-Length Video Generation, 2025.
[4] Ruhe et al., Rolling Diffusion Models, ICML 2024.

**Questions:**

- What is the empirical maximum video length that the proposed model can generate without suffering from severe quality degradation? For instance, can a model generate videos even longer than 240s?
-Why does the author choose DMD instead of other techniques, e.g., DMD2?  Is there a specific reason?

---

> ### Author Response · Authors · 2025-11-21
> **Official Comment by Authors**
>
> Thank you for these valuable comments. We address your concerns point by point below.
> - W1: Missing related works
>     - Thanks for your reminder. We include these works in the related-work discussion and clarify their relation to LongLive in Section 2 and Appendix D.
>     - AAPT transforms a pre-trained latent video diffusion model into a 1-NFE, real-time streaming generator using adversarial post-training and supports interactive control via **camera and human pose,** whereas LongLive instead uses diffusion-based distribution-matching and train-long-test-long distillation for text-driven, multi-minute generation without GAN post-training.
>     - TECO, MALT Diffusion, and Rolling Diffusion all study *long-horizon* sequence generation or prediction by designing architectures and training schemes that better preserve temporal coherence over many frames (e.g., transformer-based predictive models in TECO on navigation-style video benchmarks, memory-augmented latent transformers in MALT on UCF-101 / Kinetics-600, and sliding-window diffusion with per-frame time reparameterization in Rolling Diffusion for Kinetics-600 video prediction and other temporal data). However, they are primarily evaluated on relatively small-scale video prediction or class-conditioned generation datasets such as UCF or Kinetics, are not designed as large-scale text-to-video systems, and do not target real-time streaming generation, which makes them conceptually related but orthogonal to the real-time, text-driven long-video setting addressed by LongLive.
> - W2: LongLive might struggle in scenarios that require longer temporal dependencies — for example, when a person disappears and reappears in the same scene, or when generating long videos with multiple scene transitions.
>     - We thank the reviewer for pointing out the importance of long context for long video generation. Balancing **efficiency and long-range temporal consistency** is indeed a very challenging problem, and LongLive is intended as a first step (and, to our knowledge, an early pioneer) toward *real-time* long video generation with autoregressive models. Our current design, which uses an attention sink and short window attention, is optimized for streaming efficiency and stability, but as the reviewer correctly notes, it can be suboptimal for very long-horizon dependencies (e.g., a person disappearing and later reappearing). We will make this limitation clearer in the paper and view it as an important direction for future work. In particular, we plan to explore dynamic or adaptive frame sinks and to combine LongLive with memory-style mechanisms inspired by recent work (e.g., FramePack, FAR-like designs), so that the model can extend its temporal horizon while remaining efficient.
> - Q1: What is the empirical maximum video length?
>     - Our current implementation can generate videos up to approximately **255 seconds**, which is essentially the empirical upper limit before we hit architectural constraints. This bound is directly determined by the RoPE context length of the base model (Wan2.1). Specifically, the temporal RoPE in Wan2.1 supports a maximum index of 1024. After accounting for the VAE’s 4× temporal compression, this corresponds to 1+1023×4=4093 frames in pixel space. Under our setting of 16 FPS, this yields roughly 4093/16≈255 seconds of video. Beyond this temporal range, the positional encoding falls outside its valid index range, so the model can no longer reliably extend the sequence.
> - Q2: Why does the author choose DMD instead of other techniques, e.g., DMD2? Is there a specific reason?
>     - We thank the reviewer for this question. In fact, our framework is closer to “DMD + (partial) DMD2” rather than using vanilla DMD only. We follow DMD2’s idea of few-step distillation. However, we do not adopt the full DMD2 pipeline with GAN and real-data supervision. DMD2 relies on a large, carefully curated real-image dataset to provide an additional adversarial signal. In our video generation setting, collecting and maintaining such high-quality, text-aligned long video data for adversarial training is very demanding in practice. We therefore keep the score-matching part of DMD as our main objective and only borrow the few-step distillation idea from DMD2. We will clarify this “DMD + partial DMD2” choice and its practical motivation in the revision.

---

### Official Review · Reviewer_tdeq · 2025-10-31

**Soundness:** 3
**Presentation:** 4
**Contribution:** 3
**Rating:** 6
**Confidence:** 2

**Summary:**

This paper presents a frame-level AR framework for real-time interactive long video generation. The paper mainly focuses on efficiency design. The authors introduce a KV-recache mechanism that refreshes cached states to ensure smooth, adherent transitions when prompts change. They also propose a tuning strategy to address quality degradation in long videos, aligning training with inference. For efficiency, the model combines short window attention with a frame sink to maintain long-range consistency while accelerating generation. The resulting 1.3B parameter model achieves 20.7 FPS on an H100 GPU, supports videos up to 240 seconds, and demonstrates strong performance on VBench for both short and long video benchmarks.

**Strengths:**

1. A new KV-recache strategy is proposed for frame-level AR with multiple prompts.
2. Fairly comprehensive evluation, demonstrating the performance on different lengths of video.
3. Some efficient strategies for training and inference are proposed for long videos.

**Weaknesses:**

1. Generalization ability. Though the authors show some long video cases, the prompt still remains in the same scene. How does it handle multiple, rapid-fire prompts? More importantly, how does it respond to prompts that introduce abrupt semantic shifts (e.g., changing from "Iron Man fighting" to "a peaceful beach") versus the logical continuations shown in the examples?
2. The "train-long-test-long" strategy is a key strength, but its supervision (DMD loss) relies on a teacher model that itself was only trained on short clips. While the student model is exposed to long rollouts, the supervision for each new clip is still provided by this short-clip teacher. This may create a ceiling for true long-range coherence, as the teacher lacks the long-horizon context the student is trying to learn.
3. The ablation on time consuming about the KV cache, no KV Cache, and KV recache for different lengths of video is suggested to add to prove the effiency.
4. The ablation about frame sink strategy (first frame or others) is encouraged to add.

**Questions:**

Please see weaknesses.

---

> ### Author Response · Authors · 2025-11-21
> **Official Comment by Authors**
>
> Thank you for these valuable comments. We address your concerns point by point below.
> - W1: How does it handle multiple, rapid-fire prompts?
>     - LongLive is not limited to single-scene continuations. It supports multiple, rapid-fire prompts. As you noted, the appendix videos (such as assets/interactive/interactive-science_compressed.mp4 and assets/interactive/interactive-action_compressed.mp4) are key examples of this capability. These demonstrations show cases with significant scene changes and storyboard-like cuts. Like the from “a fuel truck erupts” to “The protagonist in a warehouse”. As you noted, the model's heavy reliance on **local frame attention** is what ensures the video remains coherent and smooth during these rapid changes. The sink frame's global-consistency role becomes less important in these moments, but it does not have a negative impact.
> - W2: The "train-long-test-long" strategy is a key strength, but its supervision (DMD loss) relies on a teacher model that itself was only trained on short clips.
>     - Our teacher is short-clip trained, but in our *train-long-test-long* setup it acts as a local critic, not a long-horizon planner. During training, LongLive rolls out its own long videos, and we then sample short clips from these rollouts for DMD. Each supervised clip is conditioned on KV cache / hidden states / sink tokens that already encode the history, so the DMD loss is still applied in a long-context state space. **This is similar to RL: even with per-step rewards, long-horizon rollouts can still improve long-term behavior.**
>
>         Empirically, although supervision comes from a short-clip teacher, the student’s long-video coherence improves significantly, showing that we are not capped by the teacher’s original short horizon.
>
>         We agree that a strong long-video teacher could further help, and our framework can directly plug in longer-horizon teachers. The main limitation is practical: there are currently very few open, high-quality, directly usable long-video base models to serve as such teachers.
>
> - W3: The ablation on time consuming about the KV cache, no KV Cache, and KV recache for different lengths of video is suggested to add to prove the efficiency.
>     - We thank the reviewer for the suggestion. We have added the profiling result in for different KV cache strategies, as shown below.
>
>
>         | Method | Background Consistency ↑ | Subject Consistency ↑ | CLIP Score ↑ | FPS |
>         | --- | --- | --- | --- | --- |
>         | No KV cache | 92.75 | 89.59 | 28.95 | 22.8 |
>         | KV cache | 94.77 | 93.69 | 25.92 | 21.9 |
>         | KV recache | 94.81 | 94.04 | 27.87 | 20.7 |
> - W4: The ablation about frame sink strategy (first frame or others) is encouraged to add.
>     - We thank the reviewer for the suggestion. We have added a ablation on the frame-sink strategy, comparing no sink versus different numbers of initial sink frames. As shown in the table below, using a small number of sink frames (3+9) slightly improves VBench scores compared to no sink (0+12), especially in consistency scores, while increasing the number of sink frames (6+6, 9+3) makes the sink frame visually dominant and causes flashback-like, less coherent long-range dynamics and some temporal flickerings. We report the VBench numbers as a table and provide qualitative comparisons in the Section G of PDF appendix.
>
>
>         | Metric | No sink (0 + 12) | First frame (3 + 9) | First 2 frames (6 + 6) | First 3 frames (9 + 3) |
>         | --- | --- | --- | --- | --- |
>         | VBench Score↑ | 83.78 | 83.98 | 79.23 | 74.43 |
>         | Consistency Score↑ | 93.30 | 94.10 | 92.22 | 54.65 |

---

### Author Response · Authors · 2025-11-21
**General Response**

Thank all reviewers for your valuable comments and suggestions. We have revised the manuscript accordingly. These revisions are highlighted in blue.

1. Added a speed profiling analysis in the KV recache ablation section.
2. Included an additional ablation study on different numbers of sink frames in Appendix G.
3. Expanded the related work section following reviewer AEHT’s suggestions.
4. Re-ran the evaluations in Tables 1 and 2 multiple times to report more stable results.
5. Added a new figure to illustrate the pipeline of our streaming long-tuning strategy.

---

### Meta-Review · Area_Chair_6Dio · 2026-01-02

**Summary:**

The three reviewers raised a ‌coherent set of critiques‌ addressing ‌technical validity, methodological robustness, and contextual positioning‌ of the work.

**Technical rigor and evaluation (Reviewers 1 & 3):‌**\
Reviewer 1 questions the generalization to diverse prompts and the fundamental limitation of short-clip supervision for long-range modeling. Reviewer 3 adds concerns about ‌evaluation stability‌ and ‌unexplained performance degradation‌ after fine-tuning. Together, they highlight needs for:\
More rigorous testing of generalization (abrupt semantic shifts).\
Additional ablations (KV cache, frame sink) to substantiate efficiency claims.\
Re-evaluation with multiple seeds and clarification of fine-tuning trade-offs.

**Methodological scope and literature (Reviewer 2):‌**\
Reviewer 2 situates the work in the broader field, pointing out ‌missing citations‌ and the ‌inherent context-window limitation‌ of the proposed attention-sink mechanism. This raises important questions about ‌long-term dependency modeling‌ and whether the approach can handle complex multi-scene or reappearance scenarios.

‌**Common theme — long-range coherence vs. efficiency trade-off:‌**\
All reviewers implicitly or explicitly question whether the proposed efficiency gains (e.g., via attention sink, KV cache optimizations) come at the cost of ‌long-range temporal coherence‌. Reviewer 1’s concern about teacher-model supervision, Reviewer 2’s note on short effective context, and Reviewer 3’s observed instability in longer intervals all touch on this core tension.

The paper presents a promising efficiency-focused approach for long video generation, but ‌must more thoroughly validate long-range coherence, contextualize contributions within the literature, and stabilize evaluation claims‌ to meet acceptance standards.

**Reviewer Concerns:**

The main concerns of three reviewers were addressed by the rebuttal.

**Reviewer Scores:**

All three reviewer would maintain their postive scores if participated fully in the discussion.

---

### Decision · Program_Chairs · 2026-01-26

Accept (Poster)